# Chagas disease screening in pregnant Latin American women: Adherence to a systematic screening protocol in a non-endemic country

Jara Llenas-García[1,2,3]*, Philip Wikman-Jorgensen[2,4], Concepción Gil-Anguita[5], Violeta Ramos-Sesma[6], Diego Torrús-Tendero[7,8,9], Raquel Martínez-Goñi[10], Mónica Romero-Nieto[2,3,11], Javier García-Abellán[2,12], María José Esteban-Giner[13], Karenina Antelo[14], María Navarro-Cots[2,15], Fernando Buñuel[2,16], Concepción Amador[5], Josefa García-García[17], Isabel Gascón[2,18], Guillermo Telenti[2,12], Encarna Fuentes-Campos[19], Ignacio Torres[20], Adelina Gimeno-Gascón[8,21], María Montserrat Ruíz-García[2,22], Miriam Navarro[23], José-Manuel Ramos-Rincón[3,7,8]

1 Internal Medicine Department, Hospital Vega Baja, Orihuela, Spain, 2 Foundation for the Promotion of Health and Biomedical Research of the Valencia Region (FISABIO), Valencia, Spain, 3 Clinical Medicine Department, University Miguel Hernández, Elche, Spain, 4 Internal Medicine Department, University Hospital San Juan de Alicante, San Juan de Alicante, Spain, 5 Internal Medicine Department, Hospital Marina Baixa, Villajoyosa, Spain, 6 Internal Medicine Service, HLA Inmaculada Hospital, Granada, Spain, 7 Reference Unit of Imported Diseases and International Health, Alicante General University Hospital, Alicante, Spain, 8 Biomedical and Health Research Institute of Alicante (ISABIAL), Alicante, Spain, 9 Parasitology Area, University Miguel Hernández, Elche, Spain, 10 Internal Medicine Department, University Hospital Vinalopó, Elche, Spain, 11 Internal Medicine Department, Elda General University Hospital, Elda, Spain, 12 Infectious Diseases Unit. Elche General University Hospital, Elche, Spain, 13 Internal Medicine Department, Hospital Virgen de los Lirios, Alcoy, Spain, 14 Internal Medicine Department, Denia Hospital, Denia, Spain, 15 Microbiology Department, Hospital Vega Baja, Orihuela, Spain, 16 Microbiology Department, University Hospital San Juan de Alicante, San Juan de Alicante, Spain, 17 Internal Medicine Department, Torrevieja University Hospital, Torrevieja, Spain, 18 Microbiology Department, Elda General University Hospital, Elda, Spain, 19 Microbiology Department, Hospital Virgen de los Lirios, Alcoy, Spain, 20 Microbiology Department, Denia Hospital, Denia, Spain, 21 Microbiology Department, Alicante General University Hospital, Alicante, Spain, 22 Microbiology Department, Elche General University Hospital, Elche, Spain, 23 Department of Public Health, Science History and Gynaecology, University Miguel Hernández, Elche, Spain

* jarallenas@gmail.com

## Abstract

### Background

Chagas disease (CD) is a chronic parasitic disease caused by *Trypanosoma cruzi* and is endemic to continental Latin America. In Spain, the main transmission route is congenital. We aimed to assess adherence to regional recommendations of universal screening for CD during pregnancy in Latin American women in the province of Alicante from 2014 to 2018.

### Methodology/Principal findings

Retrospective quality study using two data sources: 1) delivery records of Latin American women that gave birth in the 10 public hospitals of Alicante between January 2014 and December 2018; and 2) records of Chagas serologies carried out in those centers between May 2013 and December 2018. There were 3026 deliveries in Latin American women during the study period; 1178 (38.9%) underwent CD serology. Screening adherence ranged from 17.2% to 59.3% in the different health departments and was higher in Bolivian women

**Data Availability Statement:** The final dataset is fully available at GitHub, https://github.com/pwjpwj/ChagasPregnancy.

**Funding:** The author(s) received no specific funding for this work.

**Competing interests:** The authors have declared that no competing interests exist.

(48.3%). Twenty-six deliveries (2.2%) had a positive screening; CD was confirmed in 23 (2%) deliveries of 21 women. Bolivians had the highest seroprevalence (21/112; 18.7%), followed by Colombians (1/333; 0.3%) and Ecuadorians (1/348; 0.3%). Of 21 CD-positive women (19 Bolivians, 1 Colombian, 1 Ecuadorian), infection was already known in 12 (57.1%), and 9 (42.9%) had already been treated. Only 1 of the 12 untreated women (8.3%) was treated postpartum. Follow-up started in 20 of the 23 (87.0%) neonates but was completed only in 11 (47.8%); no cases of congenital transmission were detected. Among the 1848 unscreened deliveries, we estimate 43 undiagnosed cases of CD and 1 to 2 undetected cases of congenital transmission.

## Conclusions/Significance

Adherence to recommendations of systematic screening for CD in Latin American pregnant women in Alicante can be improved. Strategies to strengthen treatment of postpartum women and monitoring of exposed newborns are needed. Currently, there may be undetected cases of congenital transmission in our province.

## Author summary

Chagas disease (CD) is a neglected tropical disease endemic to Latin America. In absence of the triatomine vector in Spain, congenital (mother-to-infant) transmission is the main infection route. The Valencian Community has recommended universal screening for CD in pregnant Latin American women since 2007. In our study we analyzed adherence to that recommendation in Alicante province from 2014 to 2018, finding that it is quite low (38.9% overall, 48.3% in Bolivians) and heterogeneous between health departments. Among unscreened pregnant women during the study period, we estimate that there could be 43 undiagnosed cases of CD and 1 to 2 undetected infections in infants. We also observed very low adherence to treatment after delivery in CD-diagnosed, untreated women (8.3%), and a low rate of completed follow-up in newborns at risk of vertical infection (47.8%). We need to improve the program in order to achieve universal CD screening in Latin American (and especially Bolivian) pregnant women, to enhance CD treatment in postpartum women, and to improve monitoring in exposed newborns through a well-established notification and follow-up circuit.

## Introduction

Chagas disease (CD) is a chronic parasitic infection caused by the protozoa *Trypanosoma cruzi*, endemic in 21 countries in continental Latin America. Vector-borne transmission is the main route of contagion in endemic areas, although infection can also spread via blood transfusion, organ transplantation, orally or from mother to child[1]. Estimated prevalence is 6–7 million infected people worldwide[2], of whom 30% to 40% have or will develop organ involvement, mainly cardiomyopathy or megaviscera (megaesophagus or/and megacolon). Antiparasitic treatment with one of the two approved drugs (benznidazole and nifurtimox) is indicated for acute and congenital CD, reactivated infections, women of childbearing age and chronic disease in children. Most experts also recommend it in adult patients to avoid progression to the symptomatic phase[1]. Recent studies have shown that parasitological response to

benznidazole treatment may be as high as 82%[3,4]. Additionally, treatment has shown to prevent vertical transmission of the infection when administered before pregnancy[5,6].

Migration of chronically infected and asymptomatic people has led to the globalization of Chagas disease[7], and Spain is the non-endemic country with the highest prevalence of CD outside the Americas, with an estimated number of 65,000 affected individuals[8]. In non-endemic destination countries, non-vectorial infection, including vertical and blood-borne transmission, is the main transmission route. To control transmission, it is essential to test all pregnant women living in or migrating from endemic countries[9,10]. According to epidemiological data from Latin America, the prevalence of CD in pregnant women ranges from 0.7% to 54%, depending on nationality, history of rural residence, and mother's age[11]. Congenital *T. cruzi* infection rate ranges in different studies from 2%[12] to 13.8%[5]; a recent meta-analysis reported vertical transmission rate of 3.8% (CI95%: 2.4–5.1%) when poor quality studies were excluded[13]. A previous meta-analysis reported a higher transmission rate in endemic countries (5%) than in non-endemic ones (2.7%)[14]. Most congenital infections are asymptomatic or present with unspecific symptoms, requiring laboratory screening for detection, while a small percentage present with severe morbidity and mortality, causing hepatosplenomegaly, anemia, meningoencephalitis, and/or respiratory insufficiency. Infected newborns carry a 20% to 30% lifetime risk of chronic symptomatic CD with cardiomyopathy or digestive tract involvement[11]. However, prompt treatment with anti-parasitic therapy can achieve cure in most cases[15,16].

Serological screening in pregnant women and examination of babies born from seropositive mothers is a suitable strategy to detect and prevent congenital Chagas disease in non-endemic areas of both endemic and non-endemic countries[10,15,17,18]. In Spain, experts recommend prenatal serological screening followed by microscopic examination or polymerase chain reaction (PCR) of cord blood from infants of seropositive mothers at birth and by one month of age. Molecular tests are not well validated for congenital transmission diagnosis and can have false positives, especially in blood samples collected from umbilical cord. However, they can be considered as uptaking tests in order to prevent patient lost-to-follow-up, when (as in our region) parasitological techniques are not available or reliable[10]. For infants not diagnosed at birth, direct parasitological methods are useful for diagnosis in infants <9 months of age, while conventional serology is recommended at 9 months of age[15]. Follow-up for 9–12 months is essential as a significant proportion of cases are only detectable at a later stage[9].

In 2007, the Valencian Community protocolized systematic screening for Chagas disease during the first antenatal care consultation in women of Latin American origin[19,20]. A pilot study in the Valencian Community in 2009–2010 showed a very high adherence rate with the screening protocol, of more than 95%[21]; however, two hospital-based studies in Alicante province in the same period yielded poorer results: 40% in Elche in 2008–2012 [22] and 41.2% in Alicante General Hospital in 2008–2012 [23].

Our primary objective was to assess adherence to the recommendations of universal CD screening during pregnancy in Latin American women in the public health system of Alicante province in a more recent period, from 2014 to 2018. We also aimed to estimate the number of undetected cases of CD in pregnant women and potentially infected newborns.

## Methods

### Ethics statement

The study was performed in accordance with the ethical standards of the Declaration of Helsinki, as revised in 2013. The Alicante General University Hospital Ethics Committee of the

Valencian Healthcare Agency approved the study (ref: CEIM PI2019/08). Informed consent was not required.

## Study design and population

We performed a retrospective cross-sectional quality study on the implementation of the systematic CD screening protocol in pregnant women of Latin American origin in the Valencian Community.

In 2018, Alicante province (Fig 1) had a population of 1,838,819, including 336,902 migrants, 22,792 of whom were Latin American women [24]. Alicante province has 10 health departments (Denia, Alcoy, Marina Baixa, San Juan de Alicante, Alicante-General, Elda, Elche-General, Elche-Crevillente, Torrevieja and Orihuela), each one with a public reference hospital.

We included all Latin American women who gave birth in one of the 10 public hospitals of Alicante province between 1 January 2014 and 31 December 2018.

## Study design and data collection

We collected the delivery records for the study period in each hospital. Latin American origin was assessed using the "nationality" field in the medical record. We also gathered the records of *Trypanosoma cruzi* serologies carried out in the microbiology labs of those hospitals between 1 May 2013 and 31 December 2018. We searched for serologies from the 8 months prior to the first delivery record to include all those done at the prenatal consultation. We included serologies sampled before delivery and until 10 days post-partum. Data collection was performed retrospectively in 2019. Newborn CD status was checked at 31 December 2019.

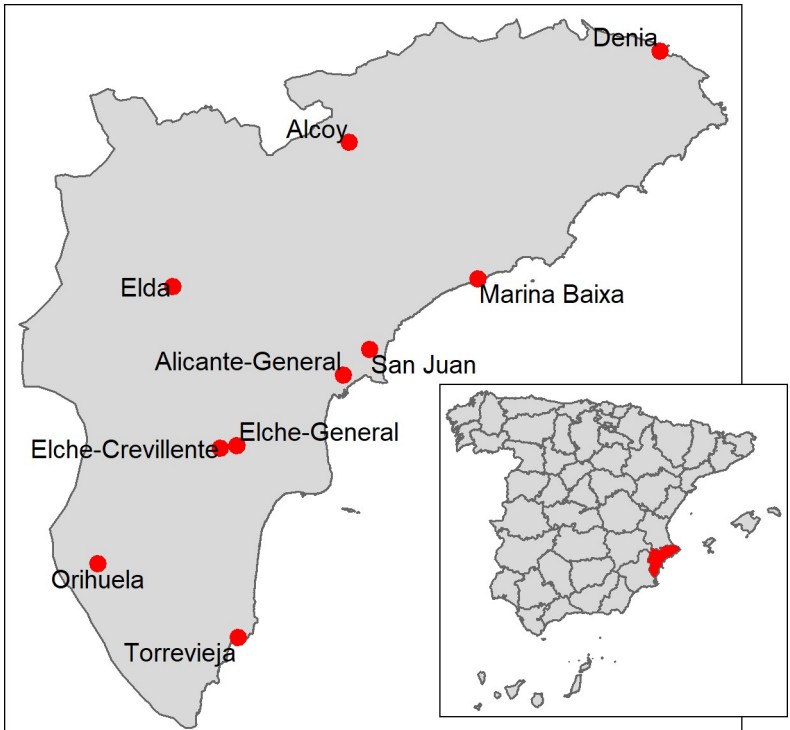

**Fig 1. Map of Spain and Alicante Province.**

We pooled the delivery records and microbiological records of nine hospitals (all but Denia) and created two respective datasets. We then cross-checked these to determine the proportion of Latin American mothers who had been screened for CD, using the unique patient identifier for the Valencian Community health card as the linking identifier. In the Denia Health Department, the health card number was not registered, so we used the medical record number to cross-reference that hospital's deliveries with its microbiological record.

The serological screening protocol differed among centers. Most used a single serological technique, such as chemiluminescence immunoassay (CLIA) (Orihuela, Elx-Crevillente, Torrevieja, Alicante- General, San Juan) or immunochromatography (Denia), while others used two different techniques (CLIA and immunochromatography in Elche-General, Elda and Marina Baixa and two different immunochromatographies in Alcoy). In those using only one serological technique, a second technique was performed only if the first one was positive (immunochromatography in Orihuela, Elx-Crevillente and Torrevieja, ELISA in Alicante-General, and San Juan, and CLIA in Denia). Discordant results were solved using with a third technique, an indirect immunofluorescence assay done in a National Reference Laboratory. CD diagnosis was confirmed by positive results from two different serological techniques using different antigens. Indeterminate results in the first serology were considered as negative.

In pregnant woman matched to a positive screening serology, medical records were reviewed and data extracted on medical history, CD confirmation serology, medical follow-up, and post-partum treatment and follow-up in the woman and her infant(s). Current protocol recommends not treating pregnant women with CD before delivery. Follow-up of the exposed newborn includes PCR ± parasitological methods at birth and at 1 month, and serology and PCR at 9–12 months[15,19]. CD workup of the positive mothers includes at least a complete medical history, physical examination, *T.cruzi* PCR, echocardiogram, electrocardiogram and a chest X-ray.

To estimate the number of missing cases we used countries' seroprevalence, as described in either a recent meta-analysis[8]or another epidemiological study[25]. To estimate the number of infected newborns, we used vertical transmission rate reported in a recent meta-analysis[13].

Sample size was not predetermined; we collected all deliveries during the study period. Some hospitals did not have electronic microbiological registries for the whole period; in those hospitals a shorter period was used (2016–2018 for San Juan de Alicante, Marina Baixa and Elche- General, and 2015–2018 for Torrevieja).

Performance and reporting of the study comply with STROBE guidelines[26].

### Statistical analysis

Categorical data are presented as absolute and relative frequencies. Prevalence was calculated with its 95% confidence interval (CI). The one-sample Kolmogorov-Smirnov test was used to assess continuous variables; as they were normally distributed, we present them as mean and standard deviation.

The chi-square test or Fisher's exact tests were used, as appropriate, to compare the distribution of categorical variables. Associations were measured using odds ratios (OR) with a 95% CI. Results were considered statistically significant if the two-tailed P value was less than 0.05. The analysis was performed using R statistical software[27].

## Results

### CD systematic screening coverage rate

There were 3026 deliveries in Latin American women during the study period; 1178 (38.9%; 95% CI 37.2%–40.7%) of the women underwent serological screening for CD. Percentage of

**Table 1. Adherence to the systematic screening protocol of Latin American pregnant women by hospital, 2014 to 2018.**

| Hospital | N deliveries | Serological techniques | Total sample | | Bolivian women | | |
| | | | N (%) serologies | N CD serologies-positive | N deliveries | N (%) serologies | N CD serologies-positive |
|---|---|---|---|---|---|---|---|
| Alicante- General | 783 | CLIA→ELISA | 464 (59.3) | 9 | 52 | 34(65.4) | 6 |
| San Juan | 203 | CLIA→ELISA | 35 (17.2) | 0 | 13 | 4 30.8) | 0 |
| Marina Baixa | 269 | CLIA+ICT | 129 (47.9) | 9 | 25 | 16 (64.0) | 9 |
| Orihuela | 255 | CLIA→ICT | 73 (28.6) | 3 | 48 | 19 (39.6) | 3 |
| Denia | 262 | ICT→CLIA | 87 (33.2) | 2 | 37 | 16 (43.2) | 1 |
| Torrevieja | 367 | CLIA→ICT | 70 (19.1) | 1 | 7 | 3 (42.9) | 0 |
| Elda | 261 | CLIA+ICT | 80 (30.6) | 0 | 15 | 5 (33.3) | 0 |
| Elche- General | 176 | CLIA+ICT | 69 (39.2) | 1 | 7 | 2 (28.6) | 1 |
| Elche-Crevillente | 330 | CLIA→ICT | 141 (42.7) | 1 | 16 | 11 (68.7) | 1 |
| Alcoy | 120 | ICT+ICT | 30 (25.0) | 0 | 12 | 2 (16.7) | 0 |
| *TOTAL* | *3026* | | *1178 (38.9)* | *26 (2.2%)* | *232* | *112 (48.3)* | *21 (18.7%)* |

CD: Chagas disease; CLIA: chemiluminescence immunoassay; ICT: immunochromatography; ELISA: enzyme-linked immunosorbent assay

pregnancies screened ranged from 59.3% (95% CI 55.8%-62.7%) in Alicante-General to 17.2% (95% CI 12.0%-22.4%) in San Juan, and it was higher in Bolivian women (48.3%; 95% CI 41.7%-54.9%; Table 1).

Fig 2 shows CD screening adherence and results over the five-year study period.

Of the 1178 serologies, 589 (50.0%) were done in the first trimester of pregnancy, 220 (18.7%) in the second trimester and 369 (31.3%) in the third trimester or first 10 days postpartum.

Thirty-six women (24.2%; 95% CI 17.3%-31.0%) from non-endemic countries, Cuba (n = 34) and the Dominican Republic (n = 2), were unnecessarily screened.

There were 3 deliveries with indeterminate results in the first serology.

## CD systematic screening results

There were 26 deliveries (2.2%) in 24 women with positive serology for CD. Of those, CD was confirmed with a second serological technique in 23 deliveries (2%, 21 women). Mean age of confirmed CD cases was 33.7 (SD 3.9) years old.

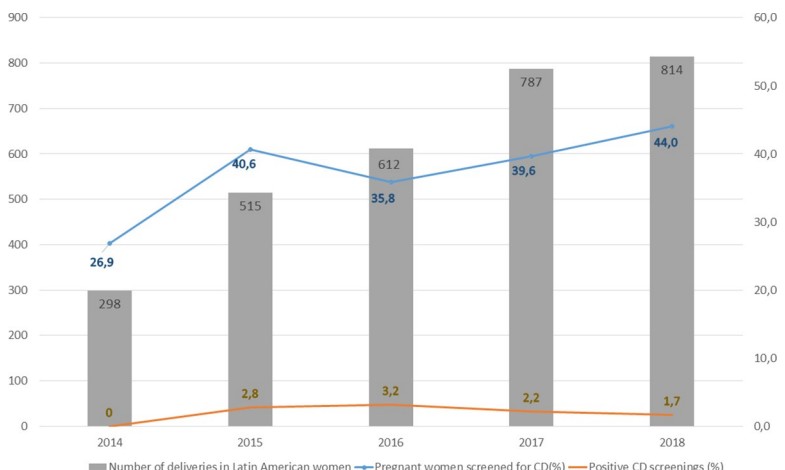

**Fig 2. Annual number of deliveries in Latin American women, percentage of pregnant Latin American women screened for Chagas disease and percentage of women with positive screening, 2014 to 2018.**

**Table 2. CD screening results in pregnant women from endemic countries according to nationality, 2014–2018.**

| Nationality of mothers | N deliveries | Deliveries screened, n (%) | Deliveries with CD-positive screen, n (%) | Deliveries with CD diagnosis, n (%) |
|---|---|---|---|---|
| Argentina | 319 | 88 (27.5) | 0 (0) | 0 (0) |
| Bolivia | 232 | 112 (48.3) | 21 (18.7) | 21 (18.7) |
| Brazil | 146 | 50 (34.2) | 0 (0) | 0 (0) |
| Chile | 29 | 3 (10.3) | 0 (0) | 0 (0) |
| Colombia | 851 | 333(39.1) | 1 (0.3) | 1 (0.3) |
| Costa Rica | 2 | 1 (50.0) | 0 (0) | 0 (0) |
| Ecuador | 807 | 348 (43.1) | 1 (0.3) | 1 (0.3) |
| El Salvador | 5 | 0 (0) | 0 (0) | 0 (0) |
| Guatemala | 7 | 4 (57.1) | 0 (0) | 0 (0) |
| Honduras | 39 | 18 (46.1) | 0 (0) | 0 (0) |
| Mexico | 33 | 9 (27.3) | 0 (0) | 0 (0) |
| Nicaragua | 15 | 8 (53.3) | 0 (0) | 0 (0) |
| Panama | 1 | 0(0) | 0 (0) | 0 (0) |
| Paraguay | 231 | 96 (41.5) | 1 (1) | 0 (0) |
| Peru | 64 | 31 (48.4) | 1 (3.2) | 0 (0) |
| Uruguay | 107 | 16 (14.9) | 0 (0) | 0 (0) |
| Venezuela | 138 | 61 (44.2) | 1 (1.6) | 0 (0) |
| *TOTAL* | *3026* | *1178 (38.9)* | *26 (2.2)* | *23 (2)* |

When analyzed by mother's nationality (Table 2), the highest seroprevalence occurred in women from Bolivia (18.7%; 95%CI 11.5%-26.0%), followed by those from Colombia (0.3%; 95% CI 0%-0.8%) and Ecuador (0.3%; 95% CI 0%-0.8%).

### Follow-up of infected mothers and their children

Of the 21 positive women (19 Bolivians, 1 Colombian, 1 Ecuadorian), 12 (57.1%) had already been diagnosed with CD, and 9 of these had already been treated. Treatment had been done a mean of 3.2 (SD 2.1) years before delivery, in most of them (8/9) with benznidazole 5mg/kg/day 60 days; only one had negative serologies in the post-treatment follow-up. Eighteen women had at least one clinical visit with a CD specialist and a workup for CD (85.7%). Of the 12 CD-positive untreated women, only one (8.3%) was treated postpartum. Twenty of the 23 newborns (87.0%) had at least one clinical visit to assess CD transmission (most of them at 1 month of age to check for at birth PCR results), but only 11 (47.8%) completed all clinical visits. No cases of congenital transmission were detected.

### Estimation of missed cases

We estimate that there would be 43 undiagnosed cases of CD among the 1848 unscreened deliveries (Table 3). Assuming a congenital transmission rate of 3.8% (95%CI: 2.4–5.1%)[13], there would be 1.6 (95%CI: 1.0–2.2) estimated cases of undetected congenital transmission.

### Discussion

The congenital CD prevention program in the Valencian Community is one of the few screening programs for controlling congenital CD in a non-endemic region and a pioneer in Spain since 2007. However, we observed adherence to this systematic screening protocol of just 38.9% in Latin American pregnant women from 2014 to 2018.

**Table 3. Estimated cases of undiagnosed CD cases among unscreened pregnant women, 2014 to 2018.**

| Nationality | Unscreened deliveries | Estimated seroprevalence (%)[8,25] | Estimated n missed cases in pregnant women |
|---|---|---|---|
| Argentina | 231 | 2.20 | 5.08 |
| Bolivia | 120 | 18.00 | 21.60 |
| Brazil | 96 | 0.60 | 0.58 |
| Chile | 26 | 1.00 | 0.26 |
| Colombia | 518 | 0.50 | 2.59 |
| Costa Rica | 1 | 0.09 | 0.00 |
| Ecuador | 459 | 0.40 | 1.84 |
| El Salvador | 5 | 3.70 | 0.19 |
| Guatemala | 3 | 0.84 | 0.03 |
| Honduras | 21 | 4.20 | 0.88 |
| Mexico | 24 | 1.50 | 0.36 |
| Nicaragua | 7 | 4.60 | 0.32 |
| Panama | 1 | 1.40 | 0.01 |
| Paraguay | 135 | 5.50 | 7.43 |
| Peru | 33 | 0.60 | 0.20 |
| Uruguay | 91 | 0.80 | 0.73 |
| Venezuela | 77 | 0.90 | 0.69 |
| *TOTAL* | *1848* | — | *42.79* |

This rate is much lower than the 95.4% reported in the pilot study in Valencia in 2009–2010 [21]. However, that was a pilot study where study midwife workshops were conducted to motivate them to screen for CD.

Our low adherence rate may be attributable to several factors. First, screening requires midwives or obstetricians to remember to add CD testing to first-trimester serologies in pregnant women from endemic countries (although serology could be made at any trimester of pregnancy in non-endemic regions). Lack of professional training on the screening program and high rates of staff turnover may have contributed to our poor rate. Workshops with midwives and automatic alerts for CD screening based on nationality in the electronic health record could improve compliance with the protocol. Including a serology at delivery in cases not previously screened could help improve screening rates. Also, Latin American women's knowledge of CD and vertical transmission is quite low[28]; increasing their knowledge and empowering women to request the proper testing could increase number of women screened [21,29]. Finally, some unscreened women may not have needed to be screened, either because they already had a CD diagnosis or had been screened earlier.

The evolution of the screening coverage rate is difficult to discern, as we only have previous data for two health departments. In Alicante-General, the coverage rate improved from 41.2% in 2008–2012 to 59.3% in 2014–2018, while in Elche-General the rate was steady (40% vs 39.2%) between 2008–2011 and 2014–2018. The absence of improvement in part might be due to the lack of an institutional program surveillance, with no feedback given to the program implementers. On the contrary, the systematic surveillance program in the region of Catalonia concluded that the congenital CD screening program was successful, with an overall coverage rate of 83.5% in 2010–15 and substantial improvements from 2010 (68.4%) to 2015 (88.6%) [30].

In our study, *T. cruzi* seroprevalence was somewhat lower than in previous studies where it ranged from 1.5% to 12% [19,30–36], probably due to the low percentage of Bolivian women included. In the largest study from Catalonia, the overall prevalence was 2.8 cases per 100

pregnancies per year but reached 15.8 in Bolivian women[30]. In our region several serological diagnostic tests are used. CLIA Architect sensitivity is reported to be 100% and specificity 96.6% and has been proposed as a single test for diagnosis[10,37]. ICT sensitivity range from 92% to 98%, while specificity is reported to be between 97% and 100%[38]. A recent meta-analysis yielded a pooled sensitivity and specificity of ICT under field conditions of 96.6% (95% CI 91.3–98.7%) and 99.3% (95% CI 98.4–99.7%), respectively[39]. ELISA and IFI have shown a sensitivity of 97–100%[38].

In our study, only one in twelve women who were candidates for postpartum treatment finally received it. Reasons for non-treatment were diverse, but loss to follow-up was the most important. High rates of lost-to-follow-up have been reported in other European programs [40] and are a threat to CD control in Europe. CD treatment is contraindicated during pregnancy to avoid potential teratogenic effects. Moreover, most guidelines recommend not treating lactating women due to concerns about benznidazole or nifurtimox transfer into breast milk[15], although some studies have found very limited transference of benznidazole or nifurtimox into breast milk[41,42]. Therefore, pregnant women diagnosed with CD were asked to come back for treatment when they stopped breastfeeding. This treatment delay may have complicated women's retention into care. Another reason for women not being treated postpartum was that the clinician wrongly considered treatment unnecessary. Better training of Spanish doctors about CD treatment recommendations is desirable[43]. In our opinion, a good referral system from the maternity clinic to a CD specialist is mandatory for the success of the program.

We are also concerned about the low rate of follow-up in newborns, also reported in other programs [44]. Integrated postpartum follow-up of mother and newborn could reduce attrition. In the Catalonian study, 82.8% of newborns were followed until serological testing at age 9–12 months. Furthermore, the program tested 34.3% of siblings, with 7.8% of those siblings being positive for CD[30], illustrating the usefulness of opportunistic screening of other family members during gestational screening programs. Indeed, a cost-efficacy study showed that the most efficient screening strategy for Latin American migrants in Spain would be to screen Latin American mothers, their newborns, and the close relatives of the mothers with a positive serology[45]. Another economic evaluation also showed the cost-effectiveness of CD active detection in pregnant women and their infants [46].

We believe that infectious, pediatric, obstetric, family and community medicine staff, plus community health workers, must work jointly to improve adherence and implement a recognized circuit that prevents the loss of CD-infected mothers and their newborns. Training sessions addressed to pediatricians and other involved health professionals would consolidate surveillance and care reference circuits, improving the control of congenital CD[47]. Affected communities' involvement is key to reduce lost-of-follow-up rates as is the creation of a comprehensive community-based CD program that includes a systematic surveillance system.

There is also a need to standardize, expand and reinforce CD screening in all women of Latin American origin who are of childbearing age (pregnant or not). Treating childbearing age women is a useful strategy to decrease vertical transmission in potential future pregnancies, since there is a correlation between the parasitemia of the mother and congenital infection[16,48,49]. We believe that a unified national regulation is necessary in order to ensure homogenous implementation of screening. Furthermore, given the worldwide dissemination of CD, a standard international program reinforcing control measures against CD transmission in non-endemic countries would be highly desirable[50]. Prenatal screening may be particularly challenging in low-prevalence settings and will require the development of innovative approaches.

Our study has several limitations. We tried to minimize losses of women screened at one health department who gave birth in another department by unifying the different registries in a single provincial database. However, we could not merge data from the hospital in Denia, so we cannot rule out that some women screened there could have delivered elsewhere. We may have also lost Latin American women giving birth at private hospitals, although we believe that there would only be a few. Latin American women may also have moved during pregnancy to or from our region, so we may have attended deliveries of women already screened in another region. However, we believe these losses were limited in number and that our data are representative of the reality of protocol implementation. Finally, there may be women already diagnosed with CD or with a previous negative serology in whom serology was not repeated.

Another limitation of the study is that we analyzed data according to nationality, not country of origin, so we may have missed women of Latin America origin who have acquired Spanish citizenship. Also, we only have data for the whole study period of six hospitals, while in the other four only data from the last years were collected, as previous electronic microbiological reports were not recorded. Country seroprevalence used to estimate missing cases could not be accurate as seroprevalence varies widely through different regions within each country. Finally, midwifery and obstetrician histories or records were not reviewed.

In conclusion, we believe that measures to boost adherence to the recommendation of systematic screening for CD in pregnant Latin American women in Alicante Province are urgently needed. The program also needs to be strengthened to improve provision of CD treatment in postpartum women and monitoring of exposed newborns through a well-established notification and follow-up circuit. Despite having a systematic CD screening program, there may be undetected cases of congenital transmission in our province.

## Acknowledgments

We'd like to acknowledge microbiologists, pediatricians, obstetricians, midwives and internal medicine specialists involved in caring for pregnant women of Latin American origin in Alicante province, as well as staff from the Documentation and Admission Department, for their kind help with this study. We are also grateful to Mundo Sano Foundation who has supported Chagas disease and *Strongyloides* screening campaigns in our province.

## Author Contributions

**Conceptualization:** Jara Llenas-García, Philip Wikman-Jorgensen, José-Manuel Ramos-Rincón.

**Data curation:** Jara Llenas-García, Philip Wikman-Jorgensen, Concepción Gil-Anguita, Violeta Ramos- Sesma, Diego Torrús-Tendero, Raquel Martínez-Goñi, Mónica Romero-Nieto, Javier García-Abellán, María José Esteban-Giner, Karenina Antelo, María Navarro-Cots, Fernando Buñuel, Concepción Amador, Josefa García-García, Isabel Gascón, Guillermo Telenti, Encarna Fuentes-Campos, Ignacio Torres, Adelina Gimeno-Gascón, María Montserrat Ruíz-García, José-Manuel Ramos-Rincón.

**Formal analysis:** Philip Wikman-Jorgensen.

**Funding acquisition:** José-Manuel Ramos-Rincón.

**Investigation:** Jara Llenas-García, Philip Wikman-Jorgensen, Concepción Gil-Anguita, Violeta Ramos- Sesma, Diego Torrús-Tendero, Raquel Martínez-Goñi, Mónica Romero-Nieto, Javier García-Abellán, María José Esteban-Giner, Karenina Antelo, María Navarro-Cots, Fernando Buñuel, Concepción Amador, Josefa García-García, Isabel Gascón, Guillermo

Telenti, Encarna Fuentes-Campos, Ignacio Torres, Adelina Gimeno-Gascón, María Montserrat Ruíz-García, Miriam Navarro, José-Manuel Ramos-Rincón.

**Methodology:** Jara Llenas-García, Philip Wikman-Jorgensen.

**Project administration:** Jara Llenas-García, José-Manuel Ramos-Rincón.

**Resources:** Jara Llenas-García, Philip Wikman-Jorgensen, Concepción Gil-Anguita, Violeta Ramos- Sesma, Diego Torrús-Tendero, Raquel Martínez-Goñi, Mónica Romero-Nieto, Javier García-Abellán, María José Esteban-Giner, Karenina Antelo, María Navarro-Cots, Fernando Buñuel, Concepción Amador, Josefa García-García, Isabel Gascón, Guillermo Telenti, Encarna Fuentes-Campos, Ignacio Torres, Adelina Gimeno-Gascón, María Montserrat Ruíz-García, José-Manuel Ramos-Rincón.

**Software:** Philip Wikman-Jorgensen.

**Supervision:** José-Manuel Ramos-Rincón.

**Validation:** Jara Llenas-García, Philip Wikman-Jorgensen.

**Visualization:** Jara Llenas-García, Philip Wikman-Jorgensen.

**Writing – original draft:** Jara Llenas-García, Philip Wikman-Jorgensen.

**Writing – review & editing:** Jara Llenas-García, Philip Wikman-Jorgensen, Concepción Gil-Anguita, Violeta Ramos- Sesma, Diego Torrús-Tendero, Raquel Martínez-Goñi, Mónica Romero-Nieto, Javier García-Abellán, María José Esteban-Giner, Karenina Antelo, María Navarro-Cots, Fernando Buñuel, Concepción Amador, Josefa García-García, Isabel Gascón, Guillermo Telenti, Encarna Fuentes-Campos, Ignacio Torres, Adelina Gimeno-Gascón, María Montserrat Ruíz-García, Miriam Navarro, José-Manuel Ramos-Rincón.

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
