## [Decision Letter · Decision Letter 0]

10 Dec 2020

Dear Ms. Llenas-García,

Thank you very much for submitting your manuscript "Chagas disease screening in pregnant Latin American women: systematic screening protocol in a non-endemic country" for consideration at PLOS Neglected Tropical Diseases. As with all papers reviewed by the journal, your manuscript was reviewed by members of the editorial board and by several independent reviewers. In light of the reviews (below this email), we would like to invite the resubmission of a significantly-revised version that takes into account the reviewers' comments. 

We cannot make any decision about publication until we have seen the revised manuscript and your response to the reviewers' comments. Your revised manuscript is also likely to be sent to reviewers for further evaluation.

Sincerely,

Erica Silberstein

Guest Editor

Alain Debrabant

Deputy Editor

Reviewer's Responses to Questions

**Key Review Criteria Required for Acceptance?**

**Methods**

-Are the objectives of the study clearly articulated with a clear testable hypothesis stated?

-Is the study design appropriate to address the stated objectives?

-Is the population clearly described and appropriate for the hypothesis being tested?

-Is the sample size sufficient to ensure adequate power to address the hypothesis being tested?

-Were correct statistical analysis used to support conclusions?

-Are there concerns about ethical or regulatory requirements being met?

Reviewer #1: see below

Reviewer #2: Line 82: `Vector-borne transmission is the main route of contagion in endemic areas”

Line 87: Please add women in childbearing age in treatment indications

Line 111: PCR of blood from umbilical cord is not a validated method for screening yet and false positive results have been reported with this technique. (Basile Luca, Ciruela Pilar, Requena-Mendez Ana, Euro Surveill. 2019;24(26):pii=1900011. https://doi.org/10.2807/1560-7917.ES.2019.24.26.19-00011) and (Buekens, P., Am. J. Trop. Med. Hyg., 98(2), 2018, pp. 478–485 doi:10.4269/ajtmh.17-0516) About infants not diagnosed at birth, you should mention the availability of parasitic direct visualization methods for diagnosis, in infants younger than 9 months of age. (Carlier et al 2019 Plos Neglect Trop Dis https://doi.org/10.1371/journal.pntd.0007694)

Line 151: It is not clear if serologic results are confirmed by two different techniques –or a third if indeterminated results. Please clarify this.

Line 155: A second technique was only performed if first was positive? Please consider that diagnosis is discharged with two different techniques being negative. (Carlier et al 2019 Plos Neglect Trop Dis https://doi.org/10.1371/journal.pntd.0007694)

Line 161: Please justify the rational for serology conducted at 1 month, as passage of mother antibodies to newborn’s blood could cause a positive serology result in not infected patients at this time. Also, serology after 9 months of age should be performed regardless previous negative or positive results. I suggest a flow chart explaining the proposed follow-up.

Line 189: Adherence to screening was not previously defined. I suggest including this definition. 

Line 219: Please detail “workup for CD”

Line 221: Please explain age of this “clinical visit” and which test was performed (PCR? Serology?) with technique description. 

Table 3. Line 228: I suggest explaining that estimated seroprevalence for each country varies widely through different regions within each country.

Reviewer #3: Llenas-Garcia et al. report adherence to screening for Chagas disease in Latin-American pregnant women in 10 health departments in Alicante province (Spain) between 2014-2018.

The study was retrospective using a cross-check between number of deliveries among LA pregnant women and serology made in the same period.

The results are interesting but there are several flawness:

1) In four departments data were available only for the period 2016-18 (San Juan, Marina Baixa, Elche General) or 2015-18 (Torrevieja) and not for the entire period of study.

2) Methods of screening were made by using different methods: single CLIA in five departments (Orihuela, Elx-Crevillente, Alicante general, San Juan, Torrevieja); one of two ICT in two depts (Alcoy, Denia); two methods (CLIA+ICT) in three depts (Marina Baixa, Elche general, Elda). Authors state that a second technique was performed if the first sample was positive. However, it is not reported what method was used nor how many samples wer not confirmed or how many indeterminate results wer observed.

3) To estimate the number of missing cases they used data gathered from two studies (Moncayo Mem Inst Oswaldo Cruz 2009; Requena-Mendez Plos Negl Trop Dis 2015); I don’t think is correct (table 3 seroprevalence for Argentina, Bolivia, Brazil, Chile, Colombia, Ecuador, El Salvador, Honduras, Mexico, Nicaragua,Uruguay, Paraguay, Venezuela from Requena-Mendez; Costa Rica, Panama, Perù from Moncaya; seroprevalence for Guatemala,?).

4) To estimate the number of infected newborns the rate of the study by Murcia (13.8%) seems too high (in a recent meta-analysis conducted in non-endemic countries was 3.5%: Colombo V et al. J Travel Med 2020 Sept 18;taaa170).

**Results**

-Does the analysis presented match the analysis plan?

-Are the results clearly and completely presented?

-Are the figures (Tables, Images) of sufficient quality for clarity?

Reviewer #1: see below

Reviewer #2: (No Response)

Reviewer #3: Other remarks:

Title: add the word “adherence” before “systematic screening protocol”

Introduction: page 5 line 84: oral transmission is not cited; page 5 line 87: two licensed drugs (is not correct the term licensed; in many countries one or both drugs are not licensed)

Line 102 cite the meta-analysis conducted in non-endemic countries (Colombo V et al. J Travel med 2020). Page 6 line 110: reference 13 is about Bolivia (the study refers to area without vectorial transmission but in an endemic country).

Study design and data collection: page 7 line 143: it is unclear why serologies are considered from May 2013

Figure 1: as it stands is unuseful; I suggest to redarwn putting a smaller map of Spain and a blow-up of the province of Alicante indicating the location of the 10 hospitals.

Table 1: add a column reporting the different methods used (CLIA, CLIA+ICT, ICT); total prevalence are not correct: 30 (1,8%) and 24 (10,4%) should be : 30 (2,3%) and 24 (18,9%).

**Conclusions**

-Are the conclusions supported by the data presented?

-Are the limitations of analysis clearly described?

-Do the authors discuss how these data can be helpful to advance our understanding of the topic under study?

-Is public health relevance addressed?

Reviewer #1: see below

Reviewer #2: Line 246: Please state that serology could be made at any trimester of pregnancy in non-endemic regions 

Line 273: According to NIH recommendations, the studies performed on breast milk with these drugs have sufficient number of patients. 

Line 283: Diagnose should be guided by WHO recommendations. Take note that PAHO/WHO consider PCR as an experimental method yet to be validated (Carlier et al 2019 Plos Neglect Trop Dis https://doi.org/10.1371/journal.pntd.0007694)

Line 286: There is evidence of false positive PCR results in neonates and this technique is still not well validated. (Basile Luca, Ciruela Pilar, Requena-Mendez Ana, Euro Surveill. 2019;24(26):pii=1900011. https://doi.org/10.2807/1560-7917.ES.2019.24.26.19-00011) (Buekens, P., Am. J. Trop. Med. Hyg., 98(2), 2018, pp. 478–485 doi:10.4269/ajtmh.17-0516)

Reviewer #3: Discussion and conclusion: Discussion is too long and several statements are not supported by the study’s results (page 13 line 246-254). The high rate of follow-up loss should be commented more deeply (it is a problem raised in many studies performed in Europe; for instance Repetto EC et al. Plos Negl Trop Dis 2015). You should focus more on your results and try to explain why no improvement was observed with previous studies in the same area (references 18 and 19).

Please comment on the different methods used to diagnose Chagas disease and their sensitivity/specificity.

Major emphasis on public health relevance should be highlightened

**Editorial and Data Presentation Modifications?**

Reviewer #1: see below

Reviewer #2: (No Response)

Reviewer #3: (No Response)

**Summary and General Comments**

Reviewer #1: The MS PNTD-D-20-01929 evaluates the implementation of the CD screening program of Latin American pregnant women in the Spanish province of Alicante between 2014 and 2018.Though such program has been improved since the previous evaluation of 2009-2010, the authors analyze the reasons of some persisting failures in order to improve it still more. Such analyze has an evident public health interest.

Our main comments concern:

1) The comparison with the previous evaluation of 2009-2010 (just mentioned l. 235), would be advantageously emphasized and discussed;

2) Since 42% of seropositive mothers were treated before their pregnancy, it would be interesting to specify the conditions of such treatment (used drug, doses, timing…) and, if possible, their serological post-treatment evolution. This is all the more important that treatment before pregnancy is recommended to avoid congenital transmission, and that no congenital infection was detected among such mothers;

3) Table 1 indicates a total of 1.8% of positive serology, whereas the data are 30/1323, i.e. 2.3%; to be corrected; 

4) The timing of sampling for serology in pregnant women should be specified.

Minor comments concern:

1) The term “vertical” transmission (including breast milk transmission) should be replaced by “congenital” or “conatal” (limited to trans-placental transmission) (keywords, l. 42, 65…);

2) References should be added, such as: PLoS NTD. 2019, 13(10), e0007694 (l. 92, 98, 107, 110, 114..); Brit J Gyn Obst 2014, 121(1), 22-33 (l. 102…); Chap 23, p517-559 in American trypanosomiasis-Chagas disease. One hundred years of research. Elsevier, 2017 ,2 edition, ISBN 978-0-12-801029-7 (l. 306); Cur Trop Med Rep 2020, 7 (4), 172-182 (l. 306)…; 

3) Table 3 should specify the source of estimated prevalences in addition to their mention in Material and Methods.

Reviewer #2: (No Response)

Reviewer #3: (No Response)

PLOS authors have the option to publish the peer review history of their article (what does this mean?). If published, this will include your full peer review and any attached files.

Reviewer #1: No

Reviewer #2: No

Reviewer #3: No
---

## [Decision Letter · Decision Letter 1]

17 Feb 2021

Dear Ms. Llenas-García,

Thank you very much for submitting your manuscript "Chagas disease screening in pregnant Latin American women: adherence to a systematic screening protocol in a non-endemic country" for consideration at PLOS Neglected Tropical Diseases. As with all papers reviewed by the journal, your manuscript was reviewed by members of the editorial board and by several independent reviewers. The reviewers appreciated the attention to an important topic. Based on the reviews, we are likely to accept this manuscript for publication, providing that you modify the manuscript according to the review recommendations. 

Please address Reviewer 1 specific request below.

Sincerely,

Erica Silberstein

Guest Editor

Alain Debrabant

Deputy Editor

Reviewer's Responses to Questions

**Key Review Criteria Required for Acceptance?**

**Methods**

-Are the objectives of the study clearly articulated with a clear testable hypothesis stated?

-Is the study design appropriate to address the stated objectives?

-Is the population clearly described and appropriate for the hypothesis being tested?

-Is the sample size sufficient to ensure adequate power to address the hypothesis being tested?

-Were correct statistical analysis used to support conclusions?

-Are there concerns about ethical or regulatory requirements being met?

Reviewer #1: (No Response)

Reviewer #2: Authors have accepted and corrected all comments.

Reviewer #3: I think that the authors have addressed in a satisfactory way the concerns raised by the reviewers

**Results**

-Does the analysis presented match the analysis plan?

-Are the results clearly and completely presented?

-Are the figures (Tables, Images) of sufficient quality for clarity?

Reviewer #1: (No Response)

Reviewer #2: Authors have accepted and corrected all comments.

Reviewer #3: The answer is yes for all the questions

Figure 1 has been redrawn in a more intelligible way.

tbales have been updated according to the requests

**Conclusions**

-Are the conclusions supported by the data presented?

-Are the limitations of analysis clearly described?

-Do the authors discuss how these data can be helpful to advance our understanding of the topic under study?

-Is public health relevance addressed?

Reviewer #1: (No Response)

Reviewer #2: Authors have accepted and corrected all comments.

Reviewer #3: Conclusions are supported by the data presented

The limitations of the study are now clearly described

The answer to the last two questions is YES

**Editorial and Data Presentation Modifications?**

Reviewer #1: Globally, we endorse the revised version of the MS PNTD-D-20-01929R1. However, we can hardly accept that the authors do not add a reference directly related to the MS topic, giving as pretext they could not read the proposed document, as mentioned:

“Reference Chap 23, p517-559 in American trypanosomiasis-Chagas disease. One hundred years of research. Elsevier, 2017 ,2 edition, ISBN 978-0-12-801029-7 has not been added because unfortunately we could not read the proposed chapter”. 

It is the duty of authors to gather all references to build their paper (it is always possible to ask the paper directly to its authors). So, publication of this MS R1 can be accepted if such reference is added (e.g. l. 109, 110, 323) (the corresponding reference paper is sent by separated email).

Reviewer #2: Authors have accepted and corrected all comments.

Reviewer #3: Trypanosoma cruzi should be in italic in the references (ref.6,12,13,14,17,20,22,30,35,47,48)

Ref 8 delete Rodrigues MM, editor

Ref 10 name of the Journal should be abbreviated, delete public library of science

Ref 13 year

Ref 14 volume

Ref 42 Chagas (Capital letter for the initial)

**Summary and General Comments**

Reviewer #1: (No Response)

Reviewer #2: Authors have accepted and corrected all comments.

Reviewer #3: The manuscript has been improved and the concerns raised by all the reviewers fully addressed

PLOS authors have the option to publish the peer review history of their article (what does this mean?). If published, this will include your full peer review and any attached files.

Reviewer #1: No

Reviewer #2: No

Reviewer #3: No

Figure Files:

Data Requirements:

Reproducibility:

References

---

## [Editor Report · Decision Letter 2]

1 Mar 2021

Dear Ms. Llenas-García,

We are pleased to inform you that your manuscript 'Chagas disease screening in pregnant Latin American women: adherence to a systematic screening protocol in a non-endemic country' has been provisionally accepted for publication in PLOS Neglected Tropical Diseases.

Best regards,

Erica Silberstein

Guest Editor

Alain Debrabant

Deputy Editor

---

## [Editor Report · Acceptance letter]

18 Mar 2021

Dear Ms. Llenas-García,

We are delighted to inform you that your manuscript, "Chagas disease screening in pregnant Latin American women: adherence to a systematic screening protocol in a non-endemic country," has been formally accepted for publication in PLOS Neglected Tropical Diseases.

Best regards,

Shaden Kamhawi

co-Editor-in-Chief

Paul Brindley

co-Editor-in-Chief
